# Ageist Attitudes Are Associated with Will-to-Live and Moderated by Age, Medical Conditions and Attitudes toward Aging

**DOI:** 10.3390/ijerph18136736

**Published:** 2021-06-23

**Authors:** Racheli-Lital Gvili, Ehud Bodner

**Affiliations:** 1Interdisciplinary Department of Social Sciences, Bar-Ilan University, Ramat-Gan 5290002, Israel; rachelygv@gmail.com; 2Department of Music, Bar-Ilan University, Ramat-Gan 5290002, Israel

**Keywords:** ageism, attitudes toward aging, medical conditions, stereotypes, will-to-live

## Abstract

The extent to which older adults’ ageist attitudes associate with their will-to-live has barely been studied. Moreover, whether this effect is moderated by older adults’ age, medical conditions, and attitudes toward their own aging has not been investigated. These associations were examined by two studies. Study 1 examined the relationship between ageist attitudes and will-to-live among individuals aged 48–97, and the moderating roles of age and medical conditions on this connection. Study 2 reassessed this connection in a new sample of older adults (people aged 60–94 years) and examined the moderating role of their attitudes toward aging in this regard. In line with the hypothesis of the first study, ageist attitudes and will-to-live were negatively associated among older adults with more medical conditions. In accordance with the hypotheses of study 2, the ageist attitudes and will-to-live connection was reconstructed, and when regressed on the ageist attitudes × attitudes toward aging interaction, it remained significant only among those with increased ageist attitudes. These findings demonstrate the negative effect that ageist attitudes may have on will-to-live, especially among the very old, and particularly when their health deteriorates, and support the utility of interventions aimed at increasing their will-to-live.

## 1. Introduction

Will-to-live (WTL) is defined as the psychological expression of one’s commitment and desire to continue living, which is a natural and a physiological instinct of all living creatures [1]. In humans, WTL is considered a marker of subjective well-being, which may outweigh the adversities of life [2]. It has been positively correlated with satisfaction with life, self-esteem, happiness, psychological prosperity, positive affect, gratitude, resilience, and meaning in life [1,3,4]. This concept also has a prognostic value in predicting survival and longevity among older adults. Several studies have demonstrated that WTL is a positive resource for older adults. For example, high levels of WTL were associated with higher survival in two longitudinal studies among Israeli women aged 70 and over, and among both men and women (aged 75–90) in Finland, even after controlling for demographics and for their medical and mental health [5,6].

While WTL is apparent throughout the life cycle, humans become more aware of it during health crises, in old age, and at the ends of their lives, as they may reassess the value of life under such threatening medical conditions and make choices which are based on their commitment and desire to continue living [7,8,9]. In this regard, WTL was found to be significantly negatively related to several aspects of coping with chronic diseases [1]. Under severe-illnesses and terminal health conditions, people with stronger WTL are more likely to prefer receiving life-sustaining treatments [10]. Unfortunately, the WTL seems to weaken with age [6,7]. Researchers explain the decrease of WTL in old age as stemming from the painful losses in extensive areas of life, which may impede one’s motivation to continue struggling during the remaining time available [11]. The aim of the current work was to examine factors which may hamper this positive resource during the second half of life and examine how it can be maintained, even among older individuals who tend to show less WTL.

Humans’ WTL is affected by values, beliefs, unique life experiences, and socio-cultural perceptions and backgrounds [1,5,7,11]. In this context, we suggest the concept of ageism, which is also subjected to such personal and socio-cultural perceptions, as a relevant factor with regard to WTL. Ageism is defined as negative stereotypes, prejudice, or discrimination against people because of their age, and is usually described in literature as directed toward older adults [12]. Ageist attitudes convey a message to older adults that they are incompetent, repulsive, and a social burden [13]. Undermining peoples’ sense of belonging and increasing their sense of being a burden for others may instigate passive and even active death wishes [14,15], which may affect WTL. Following this line of thought, we assumed that older adults with ageist attitudes may demonstrate a decreased WTL. To the best of our knowledge, only two previous studies examined the relationship between ageist stereotypes and WTL and found that WTL can be directly influenced by societally transmitted negative stereotypes about aging. In these studies, participants were subliminally primed with either negative or positive old-age stereotypes and then responded to hypothetical medical situations involving potentially fatal illnesses. The first study [16] showed that older participants who were primed with negative aging stereotypes agreed to receive fewer medical treatments when compared with those primed with positive aging stereotypes and with younger participants. The second study [17] reconstructed this effect by using a similar design and measurements of immediate and delayed age stereotypes. These results show that under life-threatening medical conditions, socially transmitted negative stereotypes of aging significantly impact older adults’ WTL and their medical decisions. Moreover, a recent qualitative study [18] conducted interviews which revealed factors influencing the WTL of Jewish adults in Israel, including societal respect for people of their own age and ageism. While societal respect for older adults was reported to be an enhancer for participants’ expressions of WTL, ageist attitudes were found to reduce expressions of WTL. These studies demonstrated a connection between priming of ageist stereotypes and WTL in the laboratory but did not examine directly how general internalized ageist attitudes affect WTL. 

Another line of research has focused on the relationship between ageist attitudes and individuals’ medical conditions. Studies found that more negative ageist attitudes predict worse physical health outcomes, such as increased hospitalization, increased stress reactivity, lower functioning and shorter longevity [19,20,21,22,23]. However, these studies did not examine the distinct effects that increased levels of medical conditions might have on older adults’ WTL. Moreover, to the best of our knowledge, there was no information regarding the possible moderating role of medical conditions in the link between ageist attitudes and WTL during the second half of life, and it is feasible that this factor would have a greater impact as individuals grow older. 

While these studies provide partial support that chronological age, medical considerations, and ageist stereotypes toward older adults may contribute to the WTL of older adults, the interactive effect of these variables concerning WTL has not been tested. It seems that understanding how these factors interact with older adults’ WTL is important, due to the critical role of WTL as a positive personal resource for older adults’ long-term survival [24]. Moreover, understanding the interactive effect of the two objective moderators (i.e., chronological age and medical conditions) on older adults’ WTL may provide support for the theoretical postulations of the stereotype embodiment theory (SET) [25]. According to the SET, people internalize negative and positive age stereotypes throughout their lives. However, as they grow older, these stereotypes begin to mold the way they perceive themselves as old people and are linked to adverse influences on their health and on their subjective well-being [10,25,26]. Following this line of thought, we assumed that older age and worse medical conditions contribute to the embodiment of these negative stereotypes about older adults, as reflected by their stronger negative effect on WTL. Therefore, the aim of the first study was to examine the moderating roles of age and medical conditions in the ageist attitudes–WTL connection among people in the second half of life. The hypothesis of the first study was that the negative association between ageist attitudes and WTL would be more pronounced among older adults with more medical conditions in comparison with older adults with fewer medical conditions and younger adults.

Following this line of thought, which proposed two objective moderators for the negative association between ageist attitudes and WTL, the aim of the second study was to examine the role of the subjective moderator of positive attitudes toward aging in mitigating the connection between ageist attitudes and WTL among older individuals. For understanding the utility of examining the interaction between ageist attitudes and attitudes toward aging, the difference between these two concepts needs to be established. While ageist attitudes refer to the social group of older adults (which respondents can perceive as their out-group), attitudes toward aging (as measured in study 2) are defined as the way older adults perceive their own aging processes [10]. Accordingly, older adults may hold to stereotypes and have prejudices against people of their age group which are irrelevant for the way they perceive themselves, as they see this group of older adults as an out-group, while still maintaining a positive view on their own aging [10,19]. While studies have shown that both negative ageist attitudes and negative attitudes toward aging have damaging effects on many aspects of older adults’ health and quality of life [21,27,28,29,30], the existing literature is lacking in studies that examine how these two different views on aging interact regarding health and well-being [19], and more specifically, WTL. 

Understanding this moderation has both theoretical and pragmatic importance. When examining the subjective moderator of positive attitudes toward aging in light of SET [25], we offer the following rationale: if this moderator mitigates the association between ageist attitudes and WTL, this may suggest that positive attitudes toward aging can assist in maintaining one’s WTL, by impeding the negative embodiment of ageist attitudes in old age. Such a finding, irrespective of its theoretical contribution, may also bear important pragmatic utility, as it will encourage interventions which focus on this cognitive aspect of attitudes toward aging. Moreover, study 2 aimed to strengthen a methodological aspect of study 1, by examining whether the ageist attitudes–WTL connection would be replicated in a different sample, consisting entirely of older adults, and by utilizing a different self-report measure of ageism. 

The hypotheses of the second study were that (1) the negative relationships between ageist attitudes and WTL would be reconstructed in the older sample and with a different ageist attitudes measure, and (2) that these relationships would be moderated by individuals’ subjective attitudes toward aging. More specifically, we hypothesized that while ageist attitudes among those with slightly positive attitudes toward aging would be negatively associated with WTL, these relationships would not be demonstrated among those with highly positive attitudes toward aging. The second hypothesis was based on the contention that while those with more ageist attitudes and positive attitudes toward aging hold negative stereotypes against others who belong to their age group, they still manage not to internalize these negative age stereotypes and perceive their own aging positively. Hence, their positive attitudes toward their own aging should balance the negative effects of strongly ageist attitudes on their WTL.

## 2. Study 1

### 2.1. Method 

#### 2.1.1. Participants and Procedure

The study examined a convenience sample of 744 community-dwelling Israeli Jews. The mean age was 68.30 (*SD* = 11.18, range: 48–97) and 53.9% were women. In terms of education, 12.9% had less than a full high school education, 33.2% had full a high school education, and 50.7% had higher education. In terms of marital status, 72.9% were married, 8.6% were divorced, 15.5% were widowers, 1.8% were living with a partner, and 1.8% were single. The participants had 3.00 children on average (*SD* = 1.53, range: 0–12). Half of them reported that they were employed (50.3%); about half (45.7%) defined their economic status as average, 21.8% as below average, and 32.5% as above average.

Research assistants approached eligible participants in their neighborhoods and large workplaces and asked them to voluntarily take part in the study. Inclusion criteria were being in their late forties (age of 48 or older), living in the community, speaking Hebrew, and having no severe cognitive impairment (assessed by asking respondents whether they were diagnosed with a severe cognitive impairment or dementia). Participants completed an online Web-based questionnaire, mostly in their homes or workplaces, which in addition to background questions and a functional impairment measure included measures of ageist attitudes, a measure of WTL, and a medical conditions questionnaire. Participants’ anonymity was kept, as they were not asked to provide their names. All signed a digital informed consent form before their participation. The study received ethical approval (number 0001) from a departmental ethical review committee in the authors’ university in accordance with the Declaration of Helsinki, on 07/11/2018. 

#### 2.1.2. Measures

Ageist attitudes were assessed by a short version [31] of the Fraboni Scale of Ageism [32]. Participants were required to rate the extent of their agreement with each of 18 statements on a scale ranging from 1 (less ageist attitudes) to 6 (more ageist attitudes). The internal validity of this scale has been demonstrated in many cultures, including Israel [31,33]. A mean score of ageism was calculated [34], and Cronbach’s alpha was 0.81. The assessment of medical conditions was done by a measure [35] based on participants’ self-reporting of 14 possible medical conditions diagnosed by a physician. The list included medical conditions such as cancer or malignant tumors; Parkinson’s disease; hip fracture or femoral fracture; heart disease or any other heart problem; stroke or cerebral vascular disease; diabetes or high blood sugar; and chronic lung diseases, such as chronic bronchitis and emphysema. A summary score of positive answers was calculated and could range between 0 and 14 (in the current study, scores ranged between 0 and 10). Will-to-live (WTL), the dependent variable [1], was measured by a single item (“In general, how would you describe your will-to-live?”), which was developed in Israel and has been used and validated in several studies concerning adults and on older adults [36]. Participants were asked to rate their answers on a scale ranging from 1 (no will at all) to 6 (very high). Functional impairment was measured by asking respondents to rate their difficulty with performing five functional activities [37], including stooping, kneeling, or crouching; reaching or extending arms above shoulder level; pulling or pushing heavy objects; lifting or carrying heavy weights; and picking up a small coin from a table. Each activity was rated on a scale ranging from 1 (not difficult to perform at all) to 4 (extremely difficult to preform). Cronbach’s α was 0.91.

#### 2.1.3. Data Analysis

After computing descriptive statistics and correlations between the study measures, we performed multiple hierarchical regression analysis, using SPSS-25, and in line with literature requirements [38,39]. Continuous variables that were used in calculating the interactions were mean centered before analyses. In order to examine the hypothesis, four steps were conducted. In step 1, WTL was regressed on covariates, which included continuous variables (number of children and functional impairment) and dichotomous variables defined as dummy variables (gender, marital status, and employment). These covariates were found to be related to WTL [4,40]. In step 2, age, number of medical conditions, and ageist attitudes were entered. In step 3, their respective three two-way interactions were entered. In step 4, their three-way interaction was entered. Significant interactions were probed and plotted using the PROCESS 3.4 version computational tool [41]. 

### 2.2. Results and Discussion

Descriptive statistics and correlations for the study variables are presented in Table 1. 

Several notable correlations can be seen in Table 1. Age was positively correlated with the number of medical conditions, with unemployment, and with functional impairment. Gender was positively correlated with marital status. Marital status was negatively correlated with functional impairment. Medical conditions were also negatively correlated with WTL, and positively correlated with unemployment and functional impairment. WTL was negatively correlated with age, with unemployment, and with functional impairment (and moderately, but significantly, with ageism).

Table 2 presents the regression analysis findings testing our hypothesis. After WTL was regressed on background variables and functional limitations (controlling for covariates), three main effects were significant—that is, age, medical conditions, and ageist attitudes. All were negatively correlated with WTL (age: *B* = −0.013, β = −0.138, *p* < 0.001; medical conditions: *B* = −0.084, β = −0.157, *p* < 0.001; ageist attitudes: *B* = −0.128, β = −0.089, *p* < 0.001). Out of the three respective two-way interactions (between age and medical conditions; between age and ageist attitudes; between ageist attitudes and medical conditions), only the ageist attitudes and medical conditions interaction was significant (*B* = −0.086, β = −0.094, *p* < 0.05). When probing this two-way interaction, it was demonstrated that while participants with more medical conditions (+1 *SD*) had a significant negative relationship between ageist attitudes and WTL (*B* = −0.275, *p* < 0.001), among participants with fewer medical conditions (−1 *SD*), the negative relationship was insignificant (*B* = −0.022, *p* = 0.771). This two-way interaction was qualified by a three-way interaction of age, medical conditions, and ageist attitudes (*B* = −0.102, β = −0.118, *p* < 0.01). 

Figure 1 presents the results of probing this interaction. The figure shows that for older participants (aged 79.5 and over) who reported higher numbers of medical conditions (between 3 and 10 conditions), high levels of ageist attitudes (between 3.32 to 4.18) were associated with reduced WTL (*B* = −0.387, *p* = 0.0001, the black dashed line). However, this association did not occur among older adults reporting few medical conditions (*B* = −0.058, *p* = 0.546, the continuous gray line, 0–1 medical conditions), nor did it occur among younger adults (aged 48 to 57.12) reporting low or high numbers of medical conditions (the dashed gray and the continuous black lines, *B* = −0.090, *p* = 0.257, and *B* = −0.026, *p* = 0.838, respectively), as they generally reported similar levels of WTL, irrespective of ageist attitudes. In other words, the levels of ageist attitudes in younger adults were not associated with their levels of WTL, whether they reported high or low numbers of medical conditions. For older adults who reported more medical conditions, the level of ageist attitudes was negatively associated with the level of WTL. Potential multicollinearity between the predicting variables was rejected, as the tolerance and VIF ratios ranges were 0.65–0.98 and 1.05–1.87, respectively, which is in line with literature requirements [42]. It should also be noted that the results of study 1, i.e., all the significant effects, remained unchanged when the study model was re-examined without the covariates. A post-hoc power analysis using G*power 3.1.9.2 [43] for multiple regressions with 12 predictors, an effect size of 0.20, and *n* = 538 yielded a power of 1.00. Therefore, the current sample size was sufficient for discovering such effect sizes.

These findings highlight the negative implications of ageist attitudes on WTL, particularly for older adults suffering from several medical conditions. Our results are in line with Levy’s stereotype embodiment theory [25], as they may imply that negative attitudes toward older adults which have been internalized over one’s life span may negatively affect one’s drive for survival when one’s medical condition is poor. The results also support existing findings in the literature [16,17], which showed that among younger adults, there were no significant effects of stereotype valence on their WTL.

## 3. Study 2

### 3.1. Method 

#### 3.1.1. Participants and Procedure

The study examined a convenience sample of 349 elderly-community-dwelling Israeli Jews. The mean age was 72.11 (*SD* = 10.12, range: 60–94) and 53.6% were women. In terms of education, 12.0% had less than a full high school education, 16.6% had a full high school education, and 71.3% had higher education. In terms of marital status, 69.1% were married, 7.7% were divorced, 14.0% were widowers, 5.4% were living with a partner, and 3.7% were single. The participants had 3.27 children on average (*SD* = 1.23, range: 1–7). Half of them reported being employed (54.7%); 24.1% defined their economic status as average, 70.8% as above average, and 5.1% as below average. Most participants reported their health as moderately good (16.0%), good (55.9%), or very good (18.6%); some reported it is not so good, (1.1%) or not good at all (8.3%).

The procedure of recruiting participants and the inclusion criteria were similar to those in the first study, with the exception of the age requirement of being 60 or older. Participants completed an online Web-based questionnaire, mostly at their homes or at their workplaces, in addition to the same background functional impairment measure. Measures also included a different assessment of ageist attitudes, a measure of attitudes toward aging, a measure of subjective health, and the same measure of WTL. Participants’ anonymity was kept, and all signed a digital informed consent form before their participation. The study received ethical approval (number 0419) by a departmental ethical review committee in the authors’ university in accordance with the Declaration of Helsinki, on 18/06/2019.

#### 3.1.2. Measures 

Ageist attitudes were measured with the Hebrew version of North and Fiske’s questionnaire [44,45]. Participants were asked to express their levels of agreement with 20 items (e.g., “Doctors spend too much time treating sickly older people”) on a scale ranging from 1 (not at all) to 5 (very much). The questionnaire includes three factors (namely succession, identity, and consumption), and a general score can be computed by averaging all items, with higher scores indicating higher ageism levels. The scale has demonstrated good divergent, convergent, and predictive validities [13]. Cronbach’s α for this study was 0.89. Attitudes to aging were examined by the short version of the Attitudes to Aging Questionnaire (AAQ) [46]. Participants were required to rate 12 items referring to negative (e.g., “I feel excluded from things because of my age”) and positive (e.g., “Growing old has been easier than I thought”) aspects of aging using a Likert scale ranging from 1 (completely disagree) to 5 (completely agree). Negative items were recoded, and all items were averaged, so that high scores reflect positive attitudes toward aging. The Hebrew version of the AAQ was previously used [47], and Cronbach’s α was 0.83. The measure of functional limitations was similar to the one used in study 1, and Cronbach’s α in the current study was 0.92. *WTL* was measured with the single item described in study 1, and background variables included the same variables used in study 1. Additionally, subjective health was assessed with a single question (“As a whole, how do you rate your health?”), rated on a scale ranging from 1 (not good at all), to 5 (very good) [48].

#### 3.1.3. Data Analysis

After creating descriptive measures and correlations matrix for the study measures, we performed two-way regression multiple hierarchical regression analysis, using SPSS-25. Continuous variables that were used in calculating the interactions were mean centered before analyses. In order to test the hypotheses, WTL was regressed on covariates which included interval scale variables (age, number of children, and functional impairment) and dichotomous variables defined as dummy variables (gender, marital status, and employment) in step 1, on ageist attitudes and attitudes toward aging in step 2, and on their interactions in step 3. The covariates selected were the same background variables and variables found to be related to WTL that were controlled in study 1 [4,40]. Significant interactions were probed for and plotted using the PROCESS 3.4 computational tool [41]. Potential multicollinearity between the predicting variables was rejected, as the tolerance and VIF ratios ranges were 0.63–0.96 and 1.05–1.57, respectively, which are in line with literature requirements [42]. 

### 3.2. Results and Discussion

Descriptive statistics and correlations for the study variables are presented in Table 3. 

Several notable correlations can be noticed in Table 3. Ageism was negatively correlated with subjective health, with attitudes toward aging and WTL. It was positively correlated with unemployment. WTL was positively correlated with attitudes toward aging and subjective health. Subjective health was positively correlated with attitudes toward aging, and negatively correlated with unemployment and functional impairment. 

Similarly to the findings of study 1, the regression analysis demonstrated that after controlling for the effect of covariates, Δ*R*^2^ = 0.15, *p* < 0.0001, ageist attitudes were associated with reduced WTL (*B* = −0.113, β = −1.05, *p* = 0.052). Moreover, positive attitudes toward aging were associated with increased WTL (*B* = 0.437, β = 0.342, *p* < 0.0001). Ageist attitudes and attitudes toward aging accounted for an additional 11% variance in WTL (*p* < 0.0001, 2% and 9% variance explained by ageist attitudes and attitudes toward aging, respectively). Moreover, the interaction between ageist attitudes and attitudes toward aging was significant (*B* = 0.096, β = 0.124, *p* = 0.014), providing an additional 1.5% variance in WTL. It should be noted that the results remained unchanged when the study model was examined without the covariates.

Upon probing the significant interaction using PROCESS, WTL was highest when ageist attitudes (both low and high) were coupled with higher positive attitudes toward aging (see Figure 2). Thus, when attitudes toward aging were + 1 *SD* above the average (i.e., higher positive attitudes), ageist attitudes were not significantly associated with WTL (*B* = −0.003, *p* = 0.9738). However, when attitudes toward aging were −1 *SD* below the average (i.e., low positive attitudes), the effect of ageist attitudes on WTL remained significant (*B* = −0.298, *p* < 0.0001). A post-hoc power analysis using G*power 3.1.9.2 [43] for multiple regressions with 10 predictors, an effect size of 0.25, and *n* = 349 yielded a power of 1.00. Therefore, the current sample size was sufficient for discovering such effect sizes.

These findings may imply that while both negative ageist attitudes and negative attitudes toward one’s own aging may impede older adults’ WTL, when their interactive effect is examined, attitudes toward aging seem to have a more important effect on older adults’ WTL. When attitudes toward aging are positive, they seem to hinder the effect of ageist attitudes on WTL. These findings comply with the conceptualization of ageism and attitudes toward aging as two distinct measures of views on aging [10]. Although ageist attitudes in old age may be relevant to older adults, they still target the age group, not the individual. In contrast, attitudes toward aging reflect one’s own perception of his/her aging, and therefore, their relevance to WTL is more pronounced.

## 4. General Discussion

The findings of the two studies illuminate the disturbing implications that ageist attitudes and negative attitudes toward aging may have on WTL in the second half of life. To the best of our knowledge, there is currently no reference in the literature to the moderating role of medical conditions in the link between ageist attitudes and WTL in the second half of life, which according to our findings becomes increasingly relevant when it is combined with old age. In this regard, study 1 emphasized a specific group of older adults, i.e., elderly-community residents with ageist attitudes who also suffered from many medical conditions and demonstrated the negative effect of these variables on their WTL. The findings of study 1 demonstrate that as people age and as their health deteriorates, their negative ageist attitudes become more strongly associated with decreased WTL. 

Study 2 also focused on the connection between ageist attitudes and WTL, but from a different angle. Its findings demonstrated an additional moderating factor which, unlike those described in study 1, was a subjective measure of the aging process. The findings of the second study seem to imply that although both negative ageist attitudes and negative attitudes toward aging impede older adults’ WTL, when they are examined together, attitudes toward aging are more relevant to their WTL. That is, when their interactive effect on WTL is measured, maintaining positive attitudes toward one’s own aging may mitigate the effect of ageist attitudes on WTL. These findings may show the ability of older adults to separate their age-based affiliations with their age groups and their views of their own aging, which has implications for their WTL. Following this interpretation, it is possible that an older person with negative attitudes toward older people, who still views his/her aging process positively (e.g., who agrees with the statement that, “Doctors spend too much time treating sickly older people,” but also with the statement, “My health is better than I expected for my age”), can succeed in maintaining his/her WTL to a higher degree. Following the argument of SET [25] that negative ageist attitudes are embodied in old age, it can be suggested that such a person can succeed in disembodying his/her negative ageist attitudes and therefore maintain the WTL. This interpretation does not imply that holding positive aging attitudes immunizes oneself against the negative effects that ageist attitudes in old age might have [21,27,28,29,30], but it does point to the importance of maintaining positive attitudes toward one’s aging in old age. 

When we planned the two studies, we designed them according to the overarching theoretical framework of SET [25]. However, when we reexamined our results, we found that they are also consistent with the stereotype threat theory [49]. According to this theory, when older people are faced with negative stereotypes about their decline [50], they feel threatened, as they fear fulfilling the nature of these stereotypes [51]. As their functioning often deteriorates due to fear, inadvertently, they affirm the stereotypes they were trying to avoid [52]. For example, the fear of older adults of having poor memory abilities can interfere with their performance in memory tests, and when the test performance is relatively poor, this confirms the stereotype that older adults lose their memory [53]. Accordingly, it has been shown that the stereotype threat has a stronger effect on older adults in general, and on older adults who identify more strongly with their old age group in particular [54,55]. In this regard, in study 1, the association between ageist attitudes and decreased WTL was significant among the older participants, particularly among older adults with several medical conditions. Accordingly, this finding may stem from the fact that these individuals felt more threatened by their advanced age and medical conditions regarding complying more with their ageist stereotypes, and subsequently, they recalled the notion that it is not worthwhile to live in old age. 

The stereotype threat theory also corresponds well with the findings of study 2. According to this theory, the threat of negative ageist attitudes towards older adults does not adversely affect those who have positive attitudes toward their own aging, and this might be because they are not frightened by negative age stereotypes, since they do not view these age stereotypes as relevant. Therefore, older adults with positive attitudes toward their own aging can hold such stereotypes and yet not exhibit the destructive stereotype threat effects on WTL.

The findings of the two studies should be interpreted in light of their methodological limitations. First, a cross-sectional design was used in both studies, thereby precluding causality. Future research should try to reconstruct our findings using longitudinal study designs. Second, while the study samples were moderate in comparison to large surveys, the sizes of the two samples were sufficient for discovering such effects. Third, this study was based on data from two convenience samples of elderly-community-dwelling older Israelis. Israel is comparable to Western societies as far as ageism is concerned [56]. However, it cannot be ruled out that certain socio-economic variables (e.g., number of children; economic status; employment status) may limit the ability to generalize the findings. While these variables were controlled in the analyses, it is nevertheless important to examine the study models in additional cultures and societies. In spite of these methodological limitations, the current study has substantive methodological strengths, as it used different tools to examine ageist attitudes in two different samples and yielded similar effects with regard to the connections between ageist attitudes and WTL.

## 5. Conclusions

Our findings demonstrate the negative impact of ageist attitudes on WTL when age increases and health deteriorates. These findings also mark the utility of positive attitudes towards aging among older adults for mitigating the negative effect of ageist attitudes on WTL. The professional literature describes a variety of interventions which elicit positive views of older adults on their aging that have been suggested in recent years [57,58,59]. Considering the positive effect that the attitudes toward ageing have on WTL [16,17,18], their potential role in mitigating the negative effect of ageist attitudes on WTL, and the positive effects that the WTL has on health and well-being [1,4], the implementation of such interventions is a promising approach for health promotion programs among older adults. The effectiveness of such interventions can be assessed by examining their impact on older adults’ WTL, as it is a unique and important indicator of well-being in old age [1,3,7] that may even predict longevity [5,6]. Additionally, longitudinal studies on the models presented in this cross-sectional are warranted. Such efforts will enrich the understanding of the effect of different views on aging on WTL and will delineate new ways for strengthening older adults’ WTL. 

## Figures and Tables

**Figure 1 ijerph-18-06736-f001:**
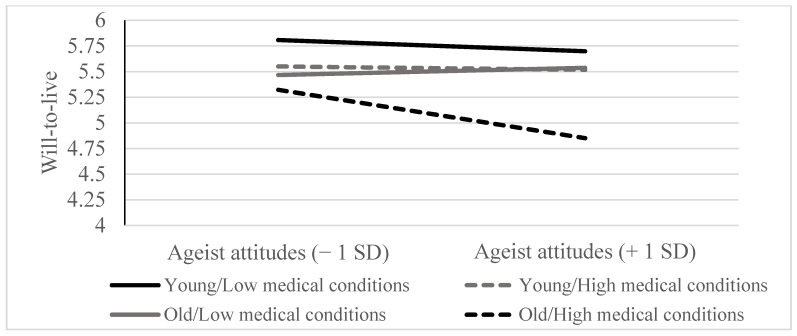
The three-way interaction between ageist attitudes, age group, and medical conditions predicting WTL.

**Figure 2 ijerph-18-06736-f002:**
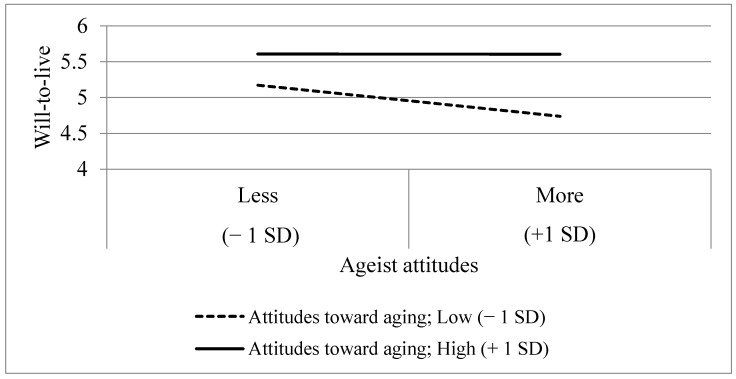
The two-way interaction between ageist attitudes and attitudes toward aging predicting WTL.

**Table 1 ijerph-18-06736-t001:** Descriptive statistics for the study (*n* = 744).

Variables	*M/%*	*SD*	1	2	3	4	5	6	7	8
1. Ageism	2.71	0.61	-							
2. MC ^a^	1.76	1.53	0.08 *	-						
3. WTL	5.43	0.82	−0.14 ***	−0.29 ***	-					
4. Age	68.30	11.81	0.13 ***	0.40 ***	−0.33 ***	-				
5. Gender ^b^	53.9%	-	0.02	0.05	0.06	0.03	-			
6. Marital status ^c^	74.1%	-	−0.02	−0.17 ***	−0.13 **	−0.26 ***	−0.26 ***	-		
7. Children	3.00	1.53	0.003	−0.01	0.06 ***	0.06	−0.07	0.17 ***	-	
8. Employment ^d^	50.3%	-	0.08 *	0.34 ***	−0.24 ***	0.66 ***	0.17 ***	−0.22 ***	0.09 *	
9. FI ^e^	1.78	1.00	0.14 **	0.40 ***	−0.34 ***	0.48	0.11 **	−0.20 ***	0.10 **	−0.35 ***

Note. Correlation values represent Pearson coefficients, except for coefficients for gender and marital status that represent point-biserial coefficients, and those for education that represent Spearman’s rank coefficients. ^a^ MC = medical conditions. ^b^ Coded: 0 = man, 1 = woman. ^c^ Coded: 0 = currently unmarried, 1 = currently married. ^d^ Coded: 0 = currently employed, 1 = currently unemployed. ^e^ FI = functional impairment. * *p* < 0.05, ** *p* < 0.01, *** *p* < 0.001.

**Table 2 ijerph-18-06736-t002:** Hierarchical linear regression predicting WTL (*n* = 538).

	B	Β	*p*
Step 1: Covariates (∆*R*^2^ = 0.140)			
Gender ^a^	0.049	0.028	0.511
Marital status ^b^	0.132	0.070	0.115
Children	−0.1	−0.02	0.604
Unemployment ^c^	−0.12	−0.07	0.121
Functional impairment	−0.26	−0.33	0.000
Step 2: Main effects (∆*R*^2^ = 0.041)			
Age	−0.013	−0.138	0.011
Medical conditions	−0.084	−0.157	0.000
Ageist attitudes	−0.128	−0.089	0.027
Step 3: Two-way interactions (∆*R*^2^ = 0.014)			
Age X Medical conditions	−0.022	−0.025	0.533
Age X Ageist attitudes	−0.045	−0.054	0.186
Ageism X Medical conditions	−0.086	−0.094	0.022
Step 4: Three-way interaction (∆*R*^2^ = 0.013)			
Age X Medical conditions X Ageist attitudes	−0.102	−0.118	0.004
*R^2^*= 0.207			

Note. ^a^ Coded: 1 = man, 2 = woman. ^b^ Coded: 1 = currently unmarried, 2 = currently married. ^c^ Coded: 0 = currently employed, 1 = currently unemployed.

**Table 3 ijerph-18-06736-t003:** Descriptive statistics for the study (*n* = 349).

Variables	*M/%*	*SD*	1	2	3	4	5	6	7	8	9
1. Ageism	2.49	0.74	-								
2. ATA ^a^	3.69	0.62	−0.24 ***	-							
3. WTL	5.24	0.79	−0.25 ***	0.46 ***	-						
4. Age	72.11	10.12	0.12 *	−0.05	−0.07	-					
5. Gender ^b^	53.6%	-	−0.04	−0.09	−0.06	0.12 *	-				
6. Marital status ^c^	74.5%	-	−0.04	0.18 **	0.10	−0.94	−0.23 ***	-			
7. Children	3.27	1.23	0.07	0.04	0.00	−0.09	−0.03	0.19 **	-		
8. Subjective health	3.82	0.87	−0.30 ***	0.44 ***	0.39 ***	0.033	−0.04	0.17 **	−0.04	-	
9. Unemployment ^d^	54.4%	-	0.29 *	−0.17 **	−0.20 ***	0.09	−0.11 *	−0.14 *	−0.03	−0.31 ***	-
10. FI ^e^	1.69	0.98	0.25 **	−0.17 **	−0.16 **	0.08	−0.13 *	−0.16 **	0.05	−0.39 ***	0.15 **

Note. Correlation values represent Pearson coefficients, except for coefficients for gender and marital status that represent point-biserial coefficients, and those for education that represent Spearman’s rank coefficients. ^a^ ATA = attitudes toward aging. ^b^ Coded: 0 = man, 1 = woman. ^c^ Coded: 0 = currently unmarried, 1 = currently married. ^d^ Coded: 0 = currently employed, 1 = currently unemployed. ^e^ FI = Functional impairment. * *p* < 0.05, ** *p* < 0.01, *** *p* < 0.001.

## Data Availability

The data that support the findings of this study are available from the corresponding author, E.B., upon reasonable request.

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
