# Peer review of "Ageist Attitudes Are Associated with Will-to-Live and Moderated by Age, Medical Conditions and Attitudes toward Aging"

_ijerph, 2021, doi:10.3390/ijerph18136736_

Round 1

Reviewer 1 Report

I am writing regarding “Ageist Attitudes are Associated with Will-to-live and Moderated by Age, Medical conditions and Attitudes Toward Aging” (ijerph-1245968). My comments follow the order of presentation of this paper, and are not in terms of importance.

Introduction

I am not sure why it is important to examine the proposed interaction effects. I do not see “has not been tested” as a strong reason.

I do not follow how the interactions effects are derived and why they should look as is.

The interaction effects are piecemeal. Why do we need to look into several moderators? Are they under the same umbrella? Study 2 looks odd. But without Study 2, the paper that would involve only Study 1 would be very weak. A stronger, overarching framework needs to be provided.

Study 1 Results

Shouldn’t all the dichotomous variables be dummy coded?

Are the small changes in R-square of concern?

Have the authors proposed two hypotheses for Study 1?

Study 2 results

Again, should dummy coding be applied?

Please clarify: “It should also be noted that the results were reconstructed without the covariates.”

General Discussion

Much content should have been presented in the Introduction to make it clearer.

Author Response

We wish to thank the Reviewer 1 for the helpful comments concerning our paper. As suggested, we implemented revisions throughout the paper and in the following lines we specify them in reference to Reviewer 1's comments. 

Reviewer: 1

Comments to the Author

First, and in line with the recommendation of Reviewer 1, the manuscript has gone extensive editing by a native English-speaking colleague.

We now turn to the Reviewer’s 1 comments.

Comment 1: I am not sure why it is important to examine the proposed interaction effects. I do not see “has not been tested” as a strong reason.

Response: We thank the Reviewer for pointing this out. In line with this comment, we now elaborate in the introduction section on the importance of the interaction effects (for Study 1 see page 2 lines 89-94 and for Study 2 see page 3 lines 123-133).

Comment 2: I do not follow how the interactions effects are derived and why they should look as is.

Response: In line with this comment, we have made several changes in the introduction section which now explain the importance of examining the effects of the moderators on the ageist attitudes-WTL connection. First, the two objective moderators of age and medical conditions in Study 1 (page 2 lines 92-94) and then, the effect of the subjective moderator of attitudes toward aging (page 3 line 106-109 and 123-133).

Comment 3: The interaction effects are piecemeal.: Why do we need to look into several moderators? Are they under the same umbrella?

Response: When presenting the results of the research model, we followed Baron and Kenny (1986) and Aiken, West and Reno (1991) recommendations for investigating and reporting a linear regression model which tests a three-way interaction. Our hypothesis focused on the three-way interaction, that is, on the interactive moderation effects of both age and medical conditions on the relationship between ageist attitudes and will-to-live. By entering the variables of the study model, step by step, to the hierarchical linear regression model we can control their individual and combined contribution to the explained variance of our outcome measure which was will-to-live. Hence, all the three potential two-way interactions were entered in the third step, and then, after considering their contribution to the explained variance of the study model, the three-way interaction was entered in the fourth step. We hope that this is now better clarified in the Study 1 subsection “data analysis” (see page 4 lines 193-201)

Comment 4: Study 2 looks odd. But without Study 2, the paper that would involve only Study 1 would be very weak. A stronger, overarching framework needs to be provided.

Response: In line with this important comment, we have changed the structure of the introduction and explained how each one of the two studies provides an examination to the same theory of SET (page 2 line 89-94 and page 3 lines 123-128). We have also emphasized the use of objective moderators in study 1 and of a subjective moderator in study 2 (page 3 lines 106-109) for the same ageist attitudes and WTL connection. Moreover, we elaborated on the methodological importance of replicating this connection with a different sample and with a different measure of ageism, which study 2 provides (page 3 lines 130-133).

Comment 5 (Study 1 Results): Shouldn’t all the dichotomous variables be dummy coded?

Response: The dichotomous variables were dummy coded, and this is now reported in the Data analysis section of Study 1 (page 4, lines 197-199).

Comment 7 (Study 1 Results): Are the small changes in R-square of concern?

Response: We agree that interactions in general, and three-way interactions in particular, often account for relatively small effect sizes. However, in the current work, the three-way interaction added 1.3% R-square change to the model (which, for such an interaction, is quite considerable), and the total R-square amounted to 20.7%, which is considered adequate (and is in line with similar study models for these concepts among older adults). Moreover, we added post-hoc power analyses for the two samples which showed that sample sizes were sufficient for discovering such effect sizes (page 7, lines 262-266 for study 1 and page 9 lines 365-367 for study 2).   

Comment 6 (Study 1 Results): Have the authors proposed two hypotheses for Study 1?

Response: No, only a three-way interaction was proposed. We thank the Reviewer for directing our attention to this unclarity which has been corrected (see page 1 line 16 and page 3 line 103).

Comment 9 (Study 2 Results): Again, should dummy coding be applied?

Response: The dichotomous variables were dummy coded, and this is now reported in the Data analysis section of Study 2 (page 8 lines 324-326).

Comment 10 (Study 2 Results): Please clarify: “It should also be noted that the results were reconstructed without the covariates.”

Response: In line with this comment the sentence was clarified (page 9, lines 354-355).

Comment 11 (General discussion): Much content should have been presented in the Introduction to make it clearer.

Response: In line with this comment, we changed the structure of the introduction, we have added information to the introduction, as explained in our responses to the previous comments. We have also cut out repetitions of previous information that has already been presented in the introduction (e.g., see the first and second paragraphs). We have reedited this section and focused on a more general discussion of our findings (e.g., page 10, lines 389-391, 409-410, 423-425, 436-443).

Finally, please see the 3 added References which were added due to the Reviewer's comments, are also marked in the manuscript  (page 12, lines 550-551, page 13, lines 552-554, lines 561-562).

Added References that are mentioned in our response to the Reviewer

  1. Aiken, L.S.; West, S.G.; Reno, R.R. Testing and probing three-way interactions. In Multiple Regression: Testing and Interpreting Interactions; Aiken, L.S., West, S.G., Reno, R.R. Eds.; Sage Publications: Newbury Park, California, 1991; pp. 49-61.
  2. Baron, R.M.; Kenny, D.A. The moderator–mediator variable distinction in social psychological research: Conceptual, strategic, and statistical considerations. Journal of Personality and Social Psychology, 1986, 51, 1173-1182. https://doi.org/10.1037//0022-3514.51.6.1173
  3. Faul, F.; Erdfelder, E.; Buchner, A.; Lang, A.G. Statistical power analyses using G* Power 3.1: Tests for correlation and re-gression analyses. Behavior Research Methods 2009, 41, 1149-1160. https://doi.org/10.3758/BRM.41.4.1149

Reviewer 2 Report

The research article Ageist Attitudes are Associated with Will-to-live and Moderated by Age, Medical conditions and Attitudes Toward Aging reflects good research. In addition, he is interesting in the field of social and psychological gerontology for his contribution to expanding the knowledge about the useful concept of will to live. Although expected, the finding that the attitudes towards aging that reflect the perception of individuals about their own aging is very relevant to the will to live is in itself useful and interesting. It is an article that in its current state is publishable with minimal modifications. The objections and comments that I make below are minor. From a methodological and statistical point of view, the research is correct and solves difficulties well. It is clear, replicable and sufficiently detailed except for one aspect that is discussed later. The support in Stereotypes embodiment theory and in Stereotype threat theory provides notable support to the research and discussion of the data carried out.

The publication could be improved if there was a more precise definition and differentiation of the concepts ageist atittudes vs. attitudes to aging that allows readers of the article a more agile reading.

Table 2 indicates that the Negative Affect has been the criterion variable to be predicted in the regression analysis. I have not been able to find any reference to it elsewhere in the article. This aspect is especially substantive since it is not described how negative affect has been evaluated in the measures section, the reason for using this criterion variable has not been justified, nor has it been related in the introduction to the relevant concepts for this article (attitudes , ageism or will to live).
It would be interesting to know the opinion of the authors when discussing the data about the generalizability of the findings. For example, in relation to the socio-demographic variables of the study. The participants have a good economic position, a number of children greater than the average in other countries (OECD = 1.65, Study = 3.27) or other aspects that could interfere with the results.

Finally, I would like to reflect a general assessment of the research that the authors might consider addressing in the discussion. The concept of will to live is undoubtedly of positive valence, it is probably the most positive psychological construct identified. It is surprising that most of the effort in the analysis of the concept of will to live has been carried out from a negative or deficit perspective, relating almost exclusively to elements of negative valence. Although Levy in a seminal investigation already pointed out that stereotypes are a moderating variable of the will to live, this should not be an obstacle to a more optimistic analysis of the concept. Recently an investigation has been published on the subject that indicates the following: “The concept of will to live is part of the theories of well-being and represents the conative component of positive attitudes towards life itself. Thus, it is conceived as an internal and stable factor that makes the person maintain a positive attitude towards life and want to continue living even in adversity situations. For this reason, the will to live is usually defined as the positive and subjective balance between the benefits and the adversities of life (Izal et al., 2018)”. This same research identifies that people with a greater will to live have higher levels of happiness, optimism, life satisfaction, psychological prosperity, positive affect, gratitude, resilience and meaning in life. It is very interesting to observe the results of this study in that attitudes toward aging are more relevant to WTL than ageism and that maintaining positive attitudes toward ones ’own aging may mitigate the effect of ageist attitudes on WTL. This comes to highlight a optimistic vision of social and psychological factors as health protectors and that the authors highlight in the last paragraph in which it suggests the promotion of positive interventions.

Author Response

We wish to thank the Reviewer 2 for the helpful comments concerning our paper. As suggested, we implemented revisions throughout the paper and in the following lines we specify them in reference to Reviewer 2's comments. 

Reviewer: 2

Comment 1: The research article Ageist Attitudes are Associated with Will-to-live and Moderated by Age, Medical conditions and Attitudes Toward Aging reflects good research. In addition, he is interesting in the field of social and psychological gerontology for his contribution to expanding the knowledge about the useful concept of will to live. Although expected, the finding that the attitudes towards aging that reflect the perception of individuals about their own aging is very relevant to the will to live is in itself useful and interesting. It is an article that in its current state is publishable with minimal modifications. The objections and comments that I make below are minor. From a methodological and statistical point of view, the research is correct and solves difficulties well. It is clear, replicable and sufficiently detailed except for one aspect that is discussed later. The support in Stereotypes embodiment theory and in Stereotype threat theory provides notable support to the research and discussion of the data carried out.

Response: We thank the Reviewer for her/his kind words.

Comment 2: The publication could be improved if there was a more precise definition and differentiation of the concepts ageist attitudes vs. attitudes to aging that allows readers of the article a more agile reading.

Response: In line with this comment, we have changed the order of the introduction. The definitions of the concepts of ageist attitudes and attitudes toward aging now clearly appear in the text (see page 3, lines 109-114).

Comment 3: Table 2 indicates that the Negative Affect has been the criterion variable to be predicted in the regression analysis. I have not been able to find any reference to it elsewhere in the article. This aspect is especially substantive since it is not described how negative affect has been evaluated in the measures section, the reason for using this criterion variable has not been justified, nor has it been related in the introduction to the relevant concepts for this article (attitudes, ageism or will to live).

Response: We thank the Reviewer for noticing this mistake which is indeed a miss-sight on our part. The caption of the template table is now corrected (see page 6 line 235).

Comment 4: It would be interesting to know the opinion of the authors when discussing the data about the generalizability of the findings. For example, in relation to the socio-demographic variables of the study. The participants have a good economic position, a number of children greater than the average in other countries (OECD = 1.65, Study = 3.27) or other aspects that could interfere with the results.

Response: We thank the Reviewer for this important comment. Following this comment, we now elaborate ono the potential demographic differences which need to be taken into account (see page 11 lines 437-443).

Comment 5: Finally, I would like to reflect a general assessment of the research that the authors might consider addressing in the discussion. The concept of will to live is undoubtedly of positive valence, it is probably the most positive psychological construct identified. It is surprising that most of the effort in the analysis of the concept of will to live has been carried out from a negative or deficit perspective, relating almost exclusively to elements of negative valence.

Response: We fully agree with this comment. In the manuscript we have emphasized the many benefits that will-to-live has, such as its contribution to a better subjective well-being and to longevity, in particular among older adults who are the subject of this investigation (see page 1 lines 28-36). Nevertheless, in order to better clarify the positive value of will-to-live we added lines to the introduction, which connects its positive qualities and our work (see page 2 lines 46-49 and 90-92). Moreover, we have made an effort in our second study to examine the effect of a personal resource (i.e., positive attitudes toward aging) that can mitigate the decrease in this valuable resource which is associated with ageist attitudes and this is now clarified in the text (see page 3 lines 125-128).

Comment 6: Although Levy in a seminal investigation already pointed out that stereotypes are a moderating variable of the will to live, this should not be an obstacle to a more optimistic analysis of the concept.

Response: We agree, please see our response to comment 5.

Comment 7: Recently an investigation has been published on the subject that indicates the following: “The concept of will to live is part of the theories of well-being and represents the conative component of positive attitudes towards life itself. Thus, it is conceived as an internal and stable factor that makes the person maintain a positive attitude towards life and want to continue living even in adversity situations. For this reason, the will to live is usually defined as the positive and subjective balance between the benefits and the adversities of life (Izal et al., 2018)”. This same research identifies that people with a greater will to live have higher levels of happiness, optimism, life satisfaction, psychological prosperity, positive affect, gratitude, resilience and meaning in life.

Response: Following this comment have added to the first paragraph two studies which included will-to-live as a measure that outweighs the adversities of life (Izal et al., 2019, see page 1 line 29) and is positively related to meaning in life (Izal et al., 2020; see page 1 line 31).

Comment 8: It is very interesting to observe the results of this study in that attitudes toward aging are more relevant to WTL than ageism and that maintaining positive attitudes toward ones ’own aging may mitigate the effect of ageist attitudes on WTL. This comes to highlight a optimistic vision of social and psychological factors as health protectors and that the authors highlight in the last paragraph in which it suggests the promotion of positive interventions.

Response: We thank the Reviewer for this positive feedback.

Added References which are in line with the Reviewer's comments (page 11, lines 466-469, page 13, lines 590-591).

  1. Bergman, Y.S.; Bodner, E.; Cohen-Fridel, S. Cross-cultural ageism: Ageism and attitudes toward aging among Jews and Arabs in Israel. International Psychogeriatrics 2013, 25(1), 6-15. https://doi.org/10.1017/S1041610212001548
  2. Izal M.; Nuevo R.; Montorio I. Successful Aging and positive psychology: Two empirically related perspectives. OBM Geriatrics2019, 3(4), 1-19. http://doi.org/10.21926/obm.geriatr.1904094
  3. Izal, M.; Bernabeu, S.; Martinez, H.; Bellot, A.; Montorio, I. Will to live as an expression of the well-being of older people. Revista Espanola de Geriatria y Gerontologia 2020, 55(2), 76-83. http://doi.org/10.1016/j.regg.2019.06.005

Round 2

Reviewer 1 Report

Good revision.